# Exploring the role of epidermal growth factor receptor variant III in meningeal tumors

**Rashmi Rana**[1]*, **Vaishnavi Rathi**[1], **Kirti Chauhan**[1], **Kriti Jain**[1], **Satnam Singh Chhabra**[2], **Rajesh Acharya**[2], **Samir Kumar Kalra**[2], **Anshul Gupta**[2], **Sunila Jain**[3], **Nirmal Kumar Ganguly**[1], **Dharmendra Kumar Yadav**[4]*

**1** Department of Research, Sir Ganga Ram Hospital, New Delhi, India, **2** Department of Neurosurgery, Sir Ganga Ram Hospital, New Delhi, India, **3** Department of Histopathology, Sir Ganga Ram Hospital, New Delhi, India, **4** Gachon Institute of Pharmaceutical Science and Department of Pharmacy, College of Pharmacy, Gachon University, Incheon, Republic of Korea

* rashmi.rana@sgrh.com (RR); dharmendra30oct@gmail.com (DKY)

**Data Availability Statement:** All relevant data are within the manuscript and its Supporting information files.

**Funding:** This study was supported by the Sir Ganga Ram Hospital (SGRH) Delhi, India and

## Abstract

Meningioma is the second most common type of intracranial brain tumor. Immunohistochemical techniques have shown prodigious results in the role of epidermal growth factor receptor variant III (EGFR vIII) in glioma and other cancers. However, the role of EGFR vIII in meningioma is still in question. This study attempt the confer searches for the position attained by EGFR vIII in progression and expression of meningioma. Immunohistochemistry technique showed that EGFR vIII is highly expressed in benign tumors as compared to the atypical meningioma with a highly significant p-value (p<0.05). Further analysis by flow cytometry results supported these findings thus presented high intensity of EGFR vIII in low grades of meningioma. The study revealed that the significant Ki 67 values, to predictor marker for survival and prognosis of the patients. Higher expression of EGFR vIII in low grades meningiomas as compared to high-grade tumors indicate towards its oncogenic properties. To our knowledge, limited studies reported in literature expressing the EGFR vIII in meningioma tumors. Hence, Opinions regarding the role that EGFR vIII in tumorigenesis and tumor progression are clearly conflicting and, therefore, it is crucial not only to find out its mechanism of action, but also to definitely identify its role in meningioma.

## Introduction

Meningiomas are predicted to constitute 13–26% of all intracranial brain tumors. They are originated from non-neuroepithelial progenitor cells and are the second most abundant tumors. Most of them are benign and asymptomatic belonging to the WHO grade I, based on histopathological classification [1]. WHO Grade II are atypical meningiomas (5–7% of all cases) and WHO Grade III is malignant meningiomas (0.17/100.000/year) [2]. The incidence rate is estimated to be 2-7/100.000/year in females and 1-5/100.000 in males [3]. The occurrence in children and adolescents is rare in both sexes [4]. Presently used therapies for the treatment of meningioma include; surgery, radiation, and stereotactic techniques [5]. Surgical therapies comprise of complete or partial resection of the tumor. However, after being

funded by the Indian Council of Medical Research (ICMR), Delhi, India (No.: VIR/24/2020/ECD-I.

**Competing interests:** The authors declare they have no conflicts of interest.

**Abbreviations:** BCA, Bicincconinic Acid Assay; BiTE, Bispecific T-cell engager; bscAbs, Bispecific antibody; DAB, 3,3'-Diaminobenzidine; DPX, Dibutylphthalate Polystyrene Xylene; EGFR vIII, Epidermal Growth Factor Receptor variant III; EGFR, Epidermal Growth Factor Receptor; FACS, Fluorescence-Activated Cell Sorting; Fc, Constant Fragment; FFPE, Formalin-Fixed Paraffin-Embedded; Fv, Variable Fragment; GBM, Glioblastoma Multiforme; HRP, Horse Reddish Peroxidase; IHC, Immunohistochemistry; IHS, Immunohistochemical Score; mAb, Monoclonal Antibody; MG, Meningioma; NSCL, Non-Small Cell Lung Cancer; P13-Akt, Phosphatidylinositol 3-kinase and Akt (protein kinase B).; PBS, Phosphate Buffer Saline; RTK, Receptor Tyrosine Kinase; SD, Standard Deviation; SI, Staining Intensity; STAT, Signal Transducers and Activators Of Transcription; WHO, World Health Organization.

removed, recurrence of the tumor is seen in both types of resection policies. The overall recurrence rate of meningiomas has been reported to be 20% approximately. Higher rates of recurrence persist in partial or incomplete resection of the tumor and more aggressive variants [6]. Diagnostic tools include imaging techniques like computerized tomography and magnetic resonance imaging. Immunohistochemical studies have been used to check the expression of various receptors. Epidermal growth factor receptor (EGFR) has been widely studied and known to overexpress in various types of cancers. Belonging to the RTK family, it regulates pathways linked with cell proliferation and differentiation of epithelial and hence, origin of tumors [7, 8].

Innumerable dysregulations in metabolism like EGFR gene amplification, mutations, and protein overexpression often alters the EGFR signaling [9]. EGFR vIII mutations (in extracellular domain of EGFR) may lead to ligand-independent receptor activation that denotes the transformed functional features [10]. However, some mutations are cancer type specific such as EGFR vI (amino-terminal deletion), EGFR vII (deletion of exons 14–15), EGFR vIII (deletion of exons 2–7). Among these variants, EGFR vII and EGFR vIII are established to be constitutively active and oncogenic [11].

Deletion of exons 2 to 7 of the EGFR gene causes an in-frame deletion of 267 amino acids from the extracellular domain of the receptor and can be found in 19% of the GBM patients [12]. EGFR vIII is an unusual tyrosine kinase that lacks a considerable part of the full-length EGFR extracellular domain. Thus, it fails bind the ligand therefore it has greater oncogenic potential when compared to the wild-type EGFR [13]. The much weaker kinase activity of EGFR vIII accounts for the increased level of tumor growth. Its expression is seen only in oncogenic cells and not in normal tissue. It is also known to be expressed in other human cancers such as breast, lung, ovarian, and medulloblastomas [14]. EGFR vIII, in combination with its unique extracellular domain (which lacks exons 2–7), possesses cancer-specific expression that fascinates many neuro-oncology researches. These features of EGFRvIII render it to be a potential target for antibody-based targeted therapies [15]. Overexpression of EGFR in meningioma is well observed but role of its mutated form in still in question [16].

The present study is an attempt to evaluate the role of EGFR vIII in meningeal tumors by immunohistochemical and flow cytometric methods. Analysis of DNA sequences and western blot for EGFR vIII were also performed to assess its role in tumor samples. However, contradictory results were obtained. The IHC findings suggest that EGFR vIII could be a part of tumorgenicity of meningiomas but other analytical techniques does not support the claim. Hence, its oncogenic role remains in questions and EGFR vIII cannot be proved as a biomarker. A comprehensive overview of EGFR vIII in Meningioma tumor as shown in Fig 1.

## Materials and methods

### Patient selection and sample collection

All tissue associated with the disease were obtained from patients during surgery, performed for the resection of the meningiomas, consecutively operated between the dates of 10.01.2018 and 31.12.2019 in the department of neurosurgery at Sir Ganga Ram Hospital-Delhi, India. These were the recently diagnosed patients whose clinical data and investigations were obtained from the clinicopathological referral sheets. Informed consent was obtained from each patient and protocols were approved by the Sir Ganga Ram Hospital, Human Ethical Committee (Ref no. EC/10/17/1270). Delhi, India. All experiments were carried out in accordance with relevant guidelines and regulations. Formalin-fixed paraffin-embedded tissues of tumors were used for immunohistochemical analysis for expression of EGFR vIII in

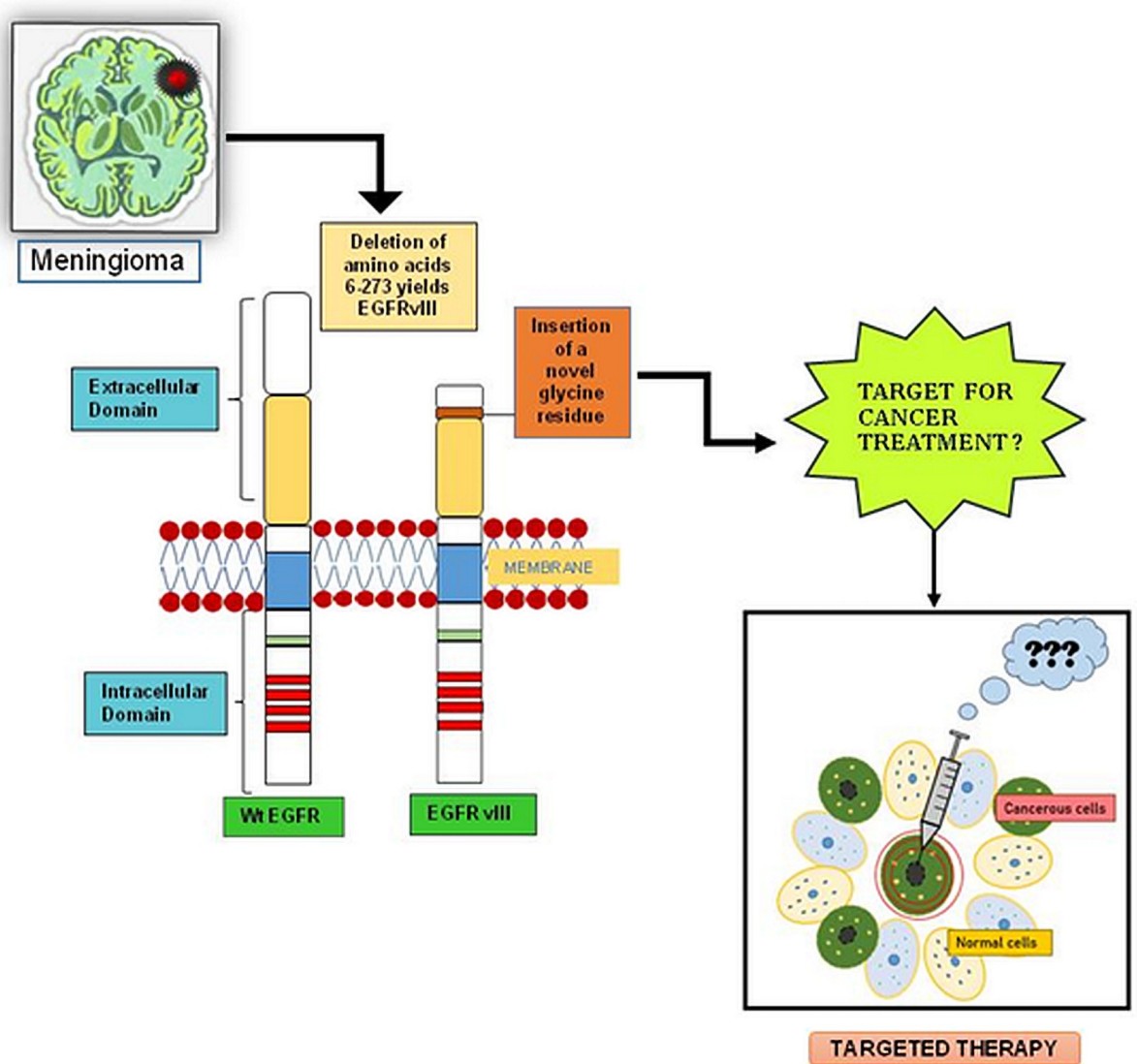

**Fig 1. The image presented here, depicts a comparative illustration of EGFR and its mutated form, EGFR VIII, formed after the deletion of 2–7 exons in the extracellular domain of parent protein molecule.** Due to this mutation, the EGFR becomes more oncogenic. However, the study given her cannot validate that in meningioma but introduce further questions about role of EGFR vIII in oncogenicity and tumorogenecity in tumor growth and formation. It also indicates towards the probable need of advanced techniques to detect the EGFR vIII in meningioma tumor. Moreover, if EGFR vIII can be used as therapeutic target is still in question.

meningioma patients. The histological sections were reviewed and all tumors were graded according to WHO criteria 2001.

*Inclusion criteria,* Patients detected with meningioma tumors were included in this study.

*Exclusion criteria,* Patients below the age of 18 years or with intraspinal or non-primary meningioma and those who were unwilling to give consent were excluded.

## Tissue specimen preparation for immunohistochemistry analysis

A section of 5-micron thickness collected on poly-L-lysine coated slides was subjected to immunohistochemistry by indirect immunoperoxidase technique. Followed by resection

tissues samples were stored in 10% formalin for fixation then embedded in the paraffin blocks. Briefly, the tissue section was cut from formalin-fixed paraffin-embedded paraffin blocks then the sections were dew axed in two changes each in xylene and alcohol and stabilized in 0.1 M phosphate buffer saline (PBS) solution. Antigen retrieval was performed by high-temperature unmasking using antigen retriever in two cycles (95˚C for 10min and at 98˚C for 5 min). Sections were then cooled at room temperature and immersed in 3% methanol hydrogen peroxide for blocking endogenous peroxidase activity. Following blockage of non-specific binding sites, protein block is used (10 min), then sections were incubated in primary antibody (anti-EGFR-vIII, clone DH8.3 monoclonal antibody and molecular weight ~140 kDa; 134.28 kDa, Merck, EMD Millipore, Temecula, CA, USA) for 1hr 30mins, followed by incubation in secondary antibody for 30 min then the addition of Streptavidin HRP (15Min). The antigen-antibody complex was visualized by using diaminobenzidine (DAB) as the chromogen. Sections were counterstained with Harris Hematoxylin, dehydrated, and mounted in DPX.

## Ki-67 proliferation marker

Histopathological grade and proliferative index determined by Ki-67 immunohistochemical expression. Immunohistochemistry for Ki-67 was carried out following the streptavidin-biotin-peroxidase method (Dako, USA) according to the protocol [17]. All immunohistochemical evaluation was performed blind concerning the clinical information. The presence or absence of brain invasion was noted in all tumors. Meningioma with brain invasion was classified as grade II meningioma.

## Immunohistochemical evaluation of EGFR vIII

Slides stained for EGFR vIII were reviewed by the observers blinded to the classification of the tumor subtype. Positive controls (placenta and endometrium) were included in each staining run. In the negative controls, the primary antibodies were omitted. The percentage of immunoreactive cells (staining percentage) were estimated by inspection and scored from 1+ to 3+. The intensity was subjectively evaluated as 1+ (weak staining), 2+ (moderate staining), or 3+ (strong staining). The EGFR vIII expression/intensity of the tumor cells was rerecorded as 1+ ($<$ 30% positive cells), 2+ ($<$60% staining), or 3+ ($>$60% staining) respectively. An immunohistochemical score (IHS) was calculated as the product of an estimate of the percentage of immunoreactive cells (staining percentage (SP) score) and the estimate of the staining intensity (staining intensity (SI) score). Analyses were conducted by the investigators on electronic images. An SI-score was calculated for each tumor and used in statistical analyses. Whole-tissue sections were evaluated using a conventional microscope (Nikon Eclipse 50i). All cases were analyzed individually by two of the authors, and discrepancies in findings were discussed and a combined consensus was attained. Each meningioma patient was given an ID unique to this study, and the investigators were consequently unaware of any case-specific clinical data during analysis.

## Flow cytometry

Total protein was isolated and quantified with the BCA method from tissue specimens of meningioma patients. 100μg of protein was suspended in phosphate buffer saline (PBS) with 5mM EDTA and incubated at RT for 15 minutes. The protein samples were treated with anti-EGFR vIII primary antibody (Merck, EMD Millipore, Temecula, CA, USA) at a dilution of 1,500 and then incubated at 4 ºC overnight on rotation. Further, treatment with Alexa Fluor 488-conjugated secondary antibody (Invitrogen) at 1, 1000 dilution and incubation at RT for 1 hour was performed. The samples were further washed with PBS containing 5 Mm EDTA.

Negative control was obtained by incubating the diluted protein sample, or in the absence of primary mAb. Protein samples for the incubations were run as duplicates, and a minimum of two experiments was done for each patient. Data were acquired in a conventional flow cytometer (FACS Aria III BD Biosciences, San Jose, CA) and analyzed with the BD Facs Diva Software (FACS Diva, version 6.6). 50000 events were recorded at a flow rate of 1.0 event per second. All antibodies were used following manufacturer instructions and protocol.

### Real time PCR

**Total RNA extraction and cDNA synthesis.** Tumor specimens were stored in -80 ˚C for upto three months. RNA extraction were prepared and examined to ensure that the tissue samples were located in representative areas of the tumors.

The RNA extracted from tumor tissue and further purity and quantity tested using Nano drop (Thermo scientific) instrument. The RNA quality was assessed by agarose electrophoresis. An aliquot of 1500ng of total RNA was reverse transcribed using High capacity cDNA Reverse Transcription Kits (applied biosystems Thermo Scientific) The reaction was conducted for 10 mins at 25ºc and then at 37ºc for 120 mins in the presence of oligo (dT) primers, and finally the enzyme was inactivated at 85 ºc for 5 mins, according to the manufacturer's instructions. The RT-PCR amplification was performed using Agilent Technologies Stratagene Mx3005P.

### Quantitative and qualitative RT-PCR

The following primers located in EGFR exons 1 and 8 were used to generate products of 92 or 893 bp for the EGFRvIII sequence, respectively: The two sets of primers (Tm: 52–60 ºC) used were: 1[st] set of EGFRvIII- Ex1-Forward: GAGTCGGGCTCTGGAGGAAA; EGFRvIII-Ex8-Reverse: CCATCTCATAGCTGTCGGGCC [18] and 2[nd] set of EGFRvIII- Ex1-Forward: GGGCTCTGGAGGAAAAGAAA; EGFRvIII- Ex8-Reverse: TGATGGAGGTGCAGTTTTTG. PCR was performed with Taq-Polymerase (Thermo) for 15 min at 95 ˚C, followed by 35 cycles for 30 sec at 95˚C, 30 sec at 60 ˚C or 58 ˚C and 1 min at 72˚C. qPCR was performed using stratagene Mx 3005P (Agilent Technologies). The analysis was performed with the use of SYBR green fluorescently labeled dye and the reference β-actin amplicons. The primers sequence sets were used following EGFRvIII-Ex1-Forward: GAGTCGGGCTCTGGAGGAAA; EGFR-vIII-Ex8-Reverse: CCATCTCATAGCTGTCGGGCC and 2[nd] set of EGFRvIII-Ex1-Forward: GGGCTCTGGAGGAAAAGAAA;EGFRvIII-Ex8-Reverse:TGATGGAGGTGCAGTTTTTG. Amplification was performed in 20-µl reaction mixture, containing cDNA amount corresponding to RNA concentration 50 ng of total RNA. The cDNA was amplified in 40 cycles: 20s denaturation at 95 ˚C, 30s annealing at 60 ˚C and 30s elongation at 72˚C. The reads were analyzed using a relative quantification (RQ) method that includes efficiency correction. The above method allowed detecting the reference β actin but EGFRvIII expression was not found in meningioma tissue specimens.

### DNA sequencing

Tumor DNA was extracted using the DNeasy Tissue Kit (Qiagen) according to the manufacturer's instructions. Genomic DNA from EGFRvIII-positive samples was used for long-range PCR to make sure the genomic deletion within intron 1 and intron 7. PCR primers and sequence primers were synthesized from the previous report with some modifications [19]. Long-range PCR was done using Platinum Taq High Fidelity DNA polymerase (Thermo) with 50 to 100 ng of genomic DNA according to the manufacturer's protocol. DNA was amplifying in 32 cycles consisting conditions, began with an initial denaturation step at 95 ˚C for 1 min,

followed by 14 cycles of 98 ˚C denaturation for 10s, 56˚C annealing for 30s and elongation at 72 ˚C for 90s. PCR products were visualized by 0.8% agarose gel electrophoresis and the break-point was confirmed by direct sequencing.

## Western blot analysis

Tumor tissue protein quantification was performed using Pierce BCA Protein Assay (Thermo Fisher Scientific, Rockford, IL) according to the manufacturer's recommended Micro plate assay procedure. Absorbance was measured with a Spectra Max M5 multi-mode microplate reader using Soft Max Pro data acquisition and analysis software (Molecular Devices, Sunnyvale CA). Tumor tissue protein were lysed in a modified lysis buffer and resuspended in denaturation buffer (4% CHAPS, 50 mM DTT, 8M Urea). Proteins were separated using 12% sodium dodecyl sulfate polyacrylamide gel electrophoresis in Tris/Glycine/SDS running buffer and transferred to Immun-Blot PVDF membrane (all reagents and supplies from Bio-Rad, Hercules, CA). Immunoblotting was performed with the following primary antibody: anti-EGFR vIII primary antibody (anti-EGFRvIII, clone DH8.3 monoclonal antibody, Merck, EMD Millipore, Temecula, CA, USA). Blots were washed and incubated with appropriate HRP-conjugated secondary antibody (A9917, Sigma-aldrich, MO, USA). Novex™ ECL Chemiluminescent Substrate Reagent Kit (invitrogen, Thermo scientific, USA) was using developing the membrane. Image was visualized in chemidoc (iBright1500, Invitrogen) and analysis software.

## Statistical analysis

All statistical testing was conducted with the statistical package for the social science system version SPSS 17.0. A P value of <0.05 was considered statistically significant. Continuous variables are presented as mean ± SD, and categorical variables are presented as absolute numbers and percentages. Association between meningioma grades and EGFR vIII intensity /expression was analyzed using the Chi-squared test. The comparison of mean Ki 67 and EGFR vIII expression between meningioma grades II and I performed using the Student's t-test.

## Results

Clinical characteristics of meningioma patients are depicted in Table 1. The tumors were originally classified according to the WHO classification 2001 [1]. Tumor tissues were formalin-fixed and embedded in paraffin. Hematoxylin/eosin-stained sections were examined microscopically to verify diagnoses and to analyze the Ki67 score. The Ki-67 protein is a cellular marker for proliferation. We demonstrated the Ki-67 labeling index level in meningioma grades I and II in Table 2. Ki-67 is the only independent predictor of both tumor recurrence and overall survival, which includes all the stages and grades under them. Ki67 blockage either by antibody microinjection or with antisense oligonucleotides leads to the capture of cell proliferation shown in Fig 2.

## Immunohistochemical analysis

Microscopic examination of the immunostained slides was done by two of the authors (RR and SJ) in collaboration using a Nikon Eclipse 50i microscope. Tissue specimens were classified based on the pathological grade and had the following distribution, benign 34/56 (60.7%) and atypical 22/56 (39.3%). Immunostained expression divided the cases into three hypothetical groups' i.e. low intensity (22/56), moderate-intensity (10/56), and strong intensity (24/56). A significant association between the intensity of EGFR VIII staining and meningioma grades

**Table 1. Patients clinical characteristics (n = 56).**

| S. No. | Variables | Frequency |
|---|---|---|
| 1 | **Age** | |
| | < 50 | 35 (19.6) |
| | ≤ 50 | 21 (11.76) |
| 2 | **Meningioma grades (n)** | |
| | I | 34 (60.7) |
| | II | 22 (39.3) |
| 3 | **Seizures** | |
| | Yes | 35 (62.5) |
| | No | 21 (37.5) |
| 4 | **Headache** | |
| | Yes | 43 (76.8) |
| | No | 13 (23.2) |
| 5 | **Vomiting** | |
| | Yes | 21 (37.5) |
| | No | 35 (62.5) |
| 6 | **Extent of resection** | |
| | (GTR, > 95%) | 53 (94.6) |
| | STR, > 85–95% | 3 (5.4) |
| 7 | **Neurological deficit** | |
| | Yes | 29 (51.8) |
| | No | 27 (48.2) |

was observed based on scores (1+, 2+, 3+) Fig 4. The majority of samples demonstrated a moderate staining intensity (SI). Generally, the atypical meningioma (grade II) exhibited a low-intensity score of EGFR vIII staining, while benign meningioma (grade I) samples demonstrated a higher intensity of staining. Specifically, 70.6% (24) of samples had intensity scores of 3+, and 29.4% (10) had intensity scores of 2+ and 0% (0) had intensity scores 1+ respectively depicted in Fig 3.

Data revealed that benign meningioma stain intensity more strongly than atypical meningioma and was statistically significant (p < 0.001) shown in Table 3 and Fig 3, 100% of atypical meningioma had EGFR vIII intensity score of 1+ shown in Table 3. Therefore, a statistically significant association between the percentage of tumor cell immunoreactivity/expression and histopathologic therapeutic (p < 0.001) is established and depicted in Table 3. The correlation between meningioma grades (benign and atypical) and EGFR vIII expression shown in Fig 4 directs that EGFR vIII can be employed as a therapeutic marker in the treatment of meningioma. The occurrence of patients only once in the Fig 5 represents that recurring meningiomas are not considered in this study.

**Table 2. Role of Ki67 marker in different grades of meningioma (Student's t test).**

| MG | N | Ki-67 (mean± SD) | P value |
|---|---|---|---|
| MG I | 34 | 4.44 1.07 | *<0.001*** |
| MGII | 22 | 9.14 2.60 | |

**Highly significant p-value<0.001.

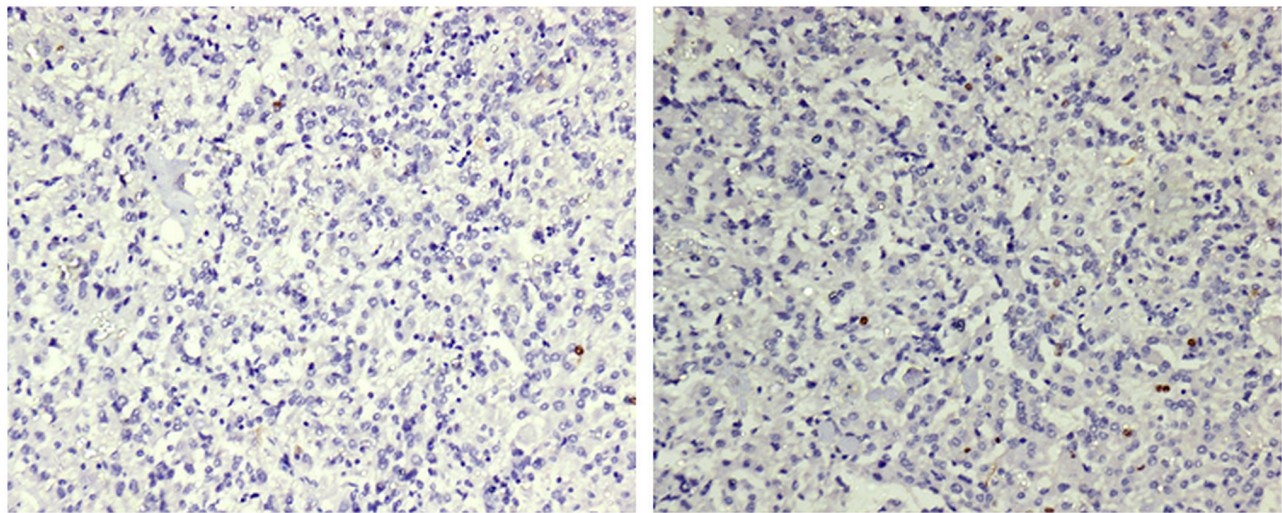

**Fig 2. These figures show the expression of Ki-67 in two different grades of meningioma (a) expression of Ki-67 in meningioma grade-I; (b) Expression of Ki-67 in meningioma grade-II.**

## Flow cytometric detection

The study for the detection of EGFR vIII included Grade I (n = 34) and Grade II (n = 22) meningioma patients. The EGFR vIII parameter staining was performed on both the sets that

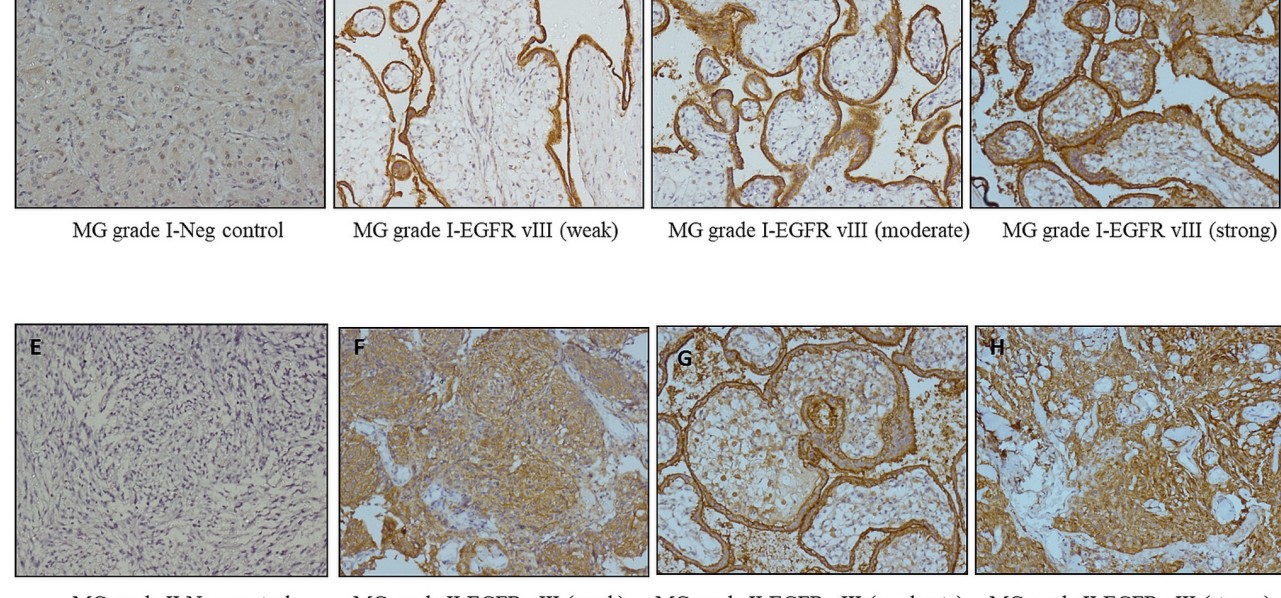

| MG grade I-Neg control | MG grade I-EGFR vIII (weak) | MG grade I-EGFR vIII (moderate) | MG grade I-EGFR vIII (strong) |
| MG grade II-Neg control | MG grade II-EGFR vIII (weak) | MG grade II-EGFR vIII (moderate) | MG grade II-EGFR vIII (strong) |

**Fig 3.** These figures show intensity scores of immunohistochemical staining **(A)** Meningioma stained with anti-EGFR showing negative staining in meningioma grade I **(B)** 1+ weak staining **(C)** 2+ moderate staining **(D)** 3+ strong staining **(E)** Meningioma stained with anti-EGFR showing negative staining in meningioma grade II **(F)** 1+ weak staining **(G)** 2+ moderate staining **(H)** 3+ strong staining. Original magnification for all images was 40 ×. Images are arranged as follows, on the left side **(A)**, **(B)**, **(C),** and **(D)** are of MG grade I. On the right side **(E)**, **(F)**, **(G),** and **(H)** belongs to meningioma grade II.

**Table 3. EGFR vIII staining intensity/expression of meningioma samples (grade I and II), n (%).**

| MG Grades | Total | EGFR VIII Intensity/Expression | | | p-value |
|---|---|---|---|---|---|
| | | 1+ | 2+ | 3+ | |
| I | 34 | 0 (0%) | 10 (29.4%) | 24 (70.6%) | <0.001** |
| II | 22 | 22 (100%) | 0 (0%) | 0 (0%) | |
| Total | 56 | 22 | 10 | 24 | |

**Highly significant p-value<0.001.

enabled us to compare the EGFR vIII expression between the two groups. Flow cytometric characterization of EGFR vIII in MG grade I and grade II are shown in Fig 5. When percentages of EGFR-positive cells after normalized subtraction were compared, a significant reduction was found in the mean value of Grade II (12.9%) as compared to the mean value of Grade I (28.2%) meningioma patients. The two groups mentioned, exhibited highly significant P values that are smaller than 0.05 Fig 2. The standard deviation between the groups observed was 8.38% and 4.34% in Grade I and Grade II meningioma patients respectively. Consequently, statistically significant flow cytometric results of EGFR vIII expression in the percentage of meningioma tumor cells for both the sets (p < 0.001) were obtained and presented in Table 4.

## EGFR variant III mRNA level in meningioma

Quantitative real-time RT-PCR revealed pronounced EGFR vIII mRNA expression that was absent in the majority of meningioma tissue specimens analyzed. The results from this analysis were in accordance with the results of the immunoblotting analysis of these tumors as shown in S1A–S1C Fig in S1 Raw images.

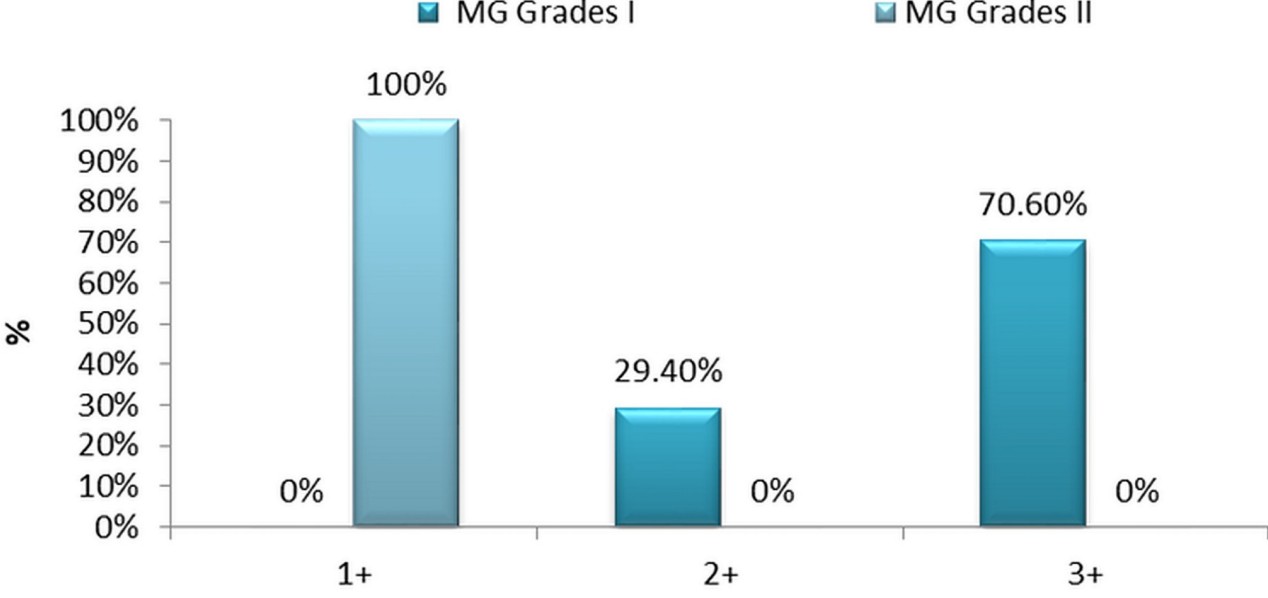

**Fig 4. Percentage of EGFR vIII staining in the human meningioma grade I and II by histopathological classification.** Tumor sections were analyzed concerning the percent of each sample exhibiting staining for EGFR vIII. The percentages of immunoreactive cells (staining percentage) were estimated by inspection and scored from +1 to +3, benign meningioma (grade I) 1+ 0% (0); 2+ 29.4% (10) and 3+ 70.6% (24) of a samples had higher intensity respectively. And atypical meningioma (grade II) were stained as follows, 1+ (weakly stained) 100% (22); 2+ (moderately stained) 0% (0); 3+ (strongly stained) 0% (0).

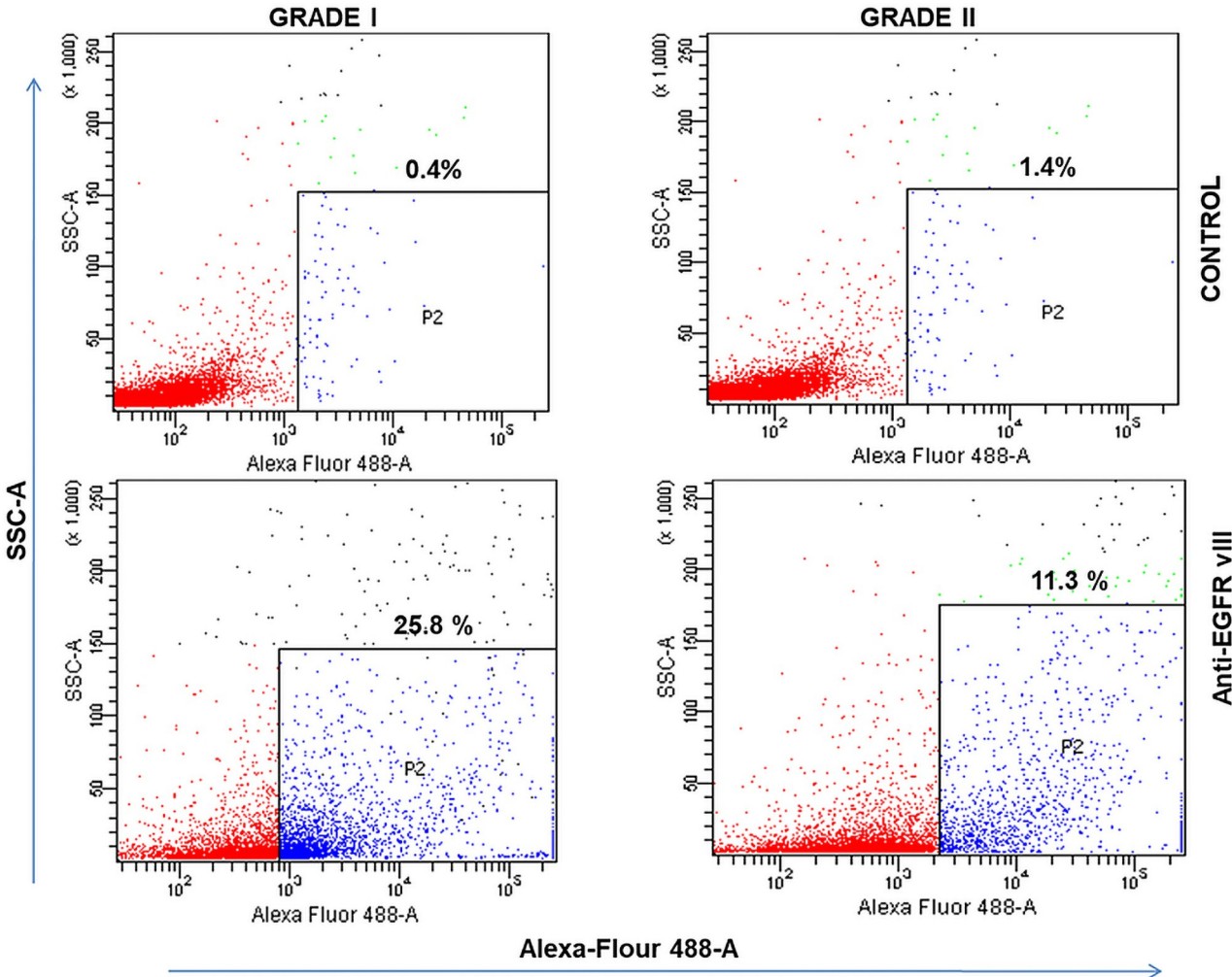

**Fig 5. Generation of EGFRvIII− and EGFRvIII+ sublines as determined via FACS.** MG Grade I representing 25.8% EGFRvIII (Alexa flour 488) positive cells (SSC vs Alexa Flour 488) while Grade II representing 11.1% EGFRvIII (Alexa flour 488) positive cells (SSC vs Alexa Flour 488). There is about 2 folds decrease in EGFR V3+ cells in grade II as compared to grade I. Unlabelled cells were used as controls in both grade I and grade II.

## DNA sequencing result

Genomic DNA was isolated from meningioma tissue specimens. We have run the gel of PCR product and found unclear appearance of bands in agrose gel then run on DNA seq for the same. Therefore, we did not found any results of *EGFRvIII in meningiomas as RT PCR results are corresponding to* DNA sequencing result shown in S2A–S2C Fig in S1 Raw images.

**Table 4. Expression of FACS EGFR VIII positive cells (%) in meningioma grade I and II patients.**

| MG Grades | FACS EGFR VIII positive cells (%) | | | p-value |
|---|---|---|---|---|
| | N | Mean | Std. Deviation | |
| 1 | 34 | 28.17% | 8.38% | <0.001 |
| 2 | 22 | 12.89% | 4.34% | |

### Identification/expression of EGFR vIII in meningioma using western blot

Protein lysates from a number of the meningiomas analyzed by Immunoblotting analysis. We did not found expression of the EGFR vIII in grades I and II tumor as shown in S3 Fig in S1 Raw images. The detection of EGFR vIII was not reflected in meningiomas specimens.

## Discussion

Meningiomas are intracranial tumors arising from the meninges of the brain and spinal cord. Despite being slow growing, they can be fatal and life threatening for suffering patients. Due to the persistent risk of relapse, there is a need to seek an alternate therapy that could be implemented to diminish the cases of recurrence [20]. Thus, the study provided tries to take step ahead towards this target. Recently several studies have been published which aims at seeking a potential target that may help in preventing and predicting their recurrence [7, 21, 22]. EGFR has already proven to be a potential target to deal with the tumors like breast, lung, and glioma, etc. whereas the capability of its variant III as a drug target is yet to be discovered. EGFR vIII is known to promote angiogenesis by activating c-myc [23] and tumor growth by activating signal transducers and activators of transcription (STAT) and P13-Akt pathways. Overexpression of EGFR is observed in 40% of the primary glioblastomas [24]. Interestingly, overexpression of EGFR vIII is observed in 50–60% of EGFR-amplified glioblastomas [25]. However, its role in the progression of the disease and survival of the patients is controversial. Some studies suggest a better prognosis of the patients with elevated EGFR vIII whereas others have found no difference in survival [26, 27]. This study does not contain the survival parameter of the patients. In this study, we determined the expression of EGFR vIII in meningiomas by immunohistochemical analysis of formalin-fixed and parafilm embedded tissues. We presented our data concerning the percent of immunoreactivity in meningioma samples and found a significant correlation between the percentage of immunoreactive staining for EGFR vIII and histopathologic grades.

Here, the study aims at assessing the role of EGFR vIII as a tumorigenic factor in meningiomas. The study was constructed to establish a correlation between EGFR vIII expression and its histopathological grades. Flow cytometric observations support the immunostaining findings. Further, statistical analysis was performed on the results obtained for their validation. We demonstrated that the intensity/expression of EGFR vIII is more prominent in lower grades of meningioma. Our data reveal a significantly greater degree of EGFR vIII expression in benign tumors as compared to atypical meningioma. Concerning the percentage of immunoreactivity, the malignant grade of meningioma demonstrated lower scores as compared to the benign and atypical meningioma. Statistical analysis proved this finding and revealed that expression of EGFR vIII is inversely correlated with tumor grades in meningioma. However, conflicting studies have also been reported which may arise from the use of variable assay methodologies. It was shown to be of no significant importance in breast cancer and does not play any role in its malignant phenotype through the EGFR wild type is positive [21]. Further, this study involves the flow cytometric analysis of the samples suspended in PBS and treated with specific primary and secondary antibodies. The immunohistochemical analysis combined with the FACS outcomes supports the probable capabilities of the EGFR vIII being an adequate marker for the diagnosis and treatment of meningioma tumors. We further, attempt to validate the data at RNA levels for corresponding sequences. However, the results obtained does not support the claim made by immunohistochemical studies. mRNA related to EGFR vIII was not detected in any meningioma tumor samples analyzed. These findings are in accordance with a previously reported study which compares the role of EGFR and EGFR vIII in meningioma and glioblastoma tumors. The authors came to the conclusion that this variant is probably not

involved in meningioma oncogenesis and progression [19]. Furthermore, a number of other studies reported the similar outcomes [28]. Additionally, expression of EGFR vIII was not observed in normal tissues that infers towards its promising candidature for targeted therapy in EGFR vIII harboring tumors, which is also supported by prior studies [29, 30]. Though it does not have any prognostic value, but its status can be helpful in glioblastoma patients in order to provide EGFR vIII targeted therapy.

Monoclonal and polyclonal antibodies aimed against EGFR vIII exhibit cross-reactivity to wild-type EGFR thus this approach of treatment impart its challenges. However, a recombinant antibody that is specific for EGFR vIII with less cross-reactivity has been developed. The antibody comprises of two anti-EGFR vIII single chains Fv's linked together and a human IgG1 Fc component [31]. Due to its high expression in cancerous cells, peptide-based targeted vaccines for its neoplasm have also been developed [32]. An immunotherapeutic strategy based on the adoptive transfer of genetically modified T-cells redirected to destroy EGFR vIII in glioblastoma cases has also been developed [33]. A study reported preclinical assessment of a bispecific antibody (bscAbs) i.e. bispecific T-cell engager (BiTE), bscEGFRvIIIxCD3 which showed that it activates T cells to mediate potent and antigen-specific lysis of EGFR vIII. Administration of this yielded prolonged survival in mice with well-developed intracerebral tumors and complete cure at rates up to 75% was achieved [34–47]. A novel monoclonal antibody (mAb) known as D2C7 reacts with both the wild-type epidermal growth factor receptor and (EGFRvIII) (both of which are major glioblastoma driver oncogenes) overexpressed on the surface of cancer cells. This immunotoxin induces secondary immune responses through the activation of T cells with being inherent tumoricidal [48–60]. Development of these types of strategies and treatment approaches are required in meningioma.

Through this study, we indicate the antagonistic expression of the EGFR vIII as compared to the EGFR in meningiomas. Although, further studies and strong evidences are required to claim this fact as validating studies gave contradictory results. We further plan to explore EGFR vIII, its expressions, and behavior in different types of tumors. The association of EGFR vIII and meningioma grade is a potential new avenue for therapeutic intervention, either as adjuvant treatment or in combination with radiation therapy. Additional clinical studies will be needed before EGFR vIII can be incorporated into clinical practice. This could be expected to substantially expand further shortly.

## Conclusion

The study could not validate the findings obtained in the preliminary assessment. Thus, a significant conclusion cannot be drawn from this analysis. Rather, the study poses numerous questions on the conventional findings and demands attention towards oncogenecity of EGFR vIII in meningioma. This study surely point towards the potentialities of the variant in number of tumors but cannot validate the results by conventional methods. Hence, further advanced techniques could be helpful in this regard. Moreover, it can also be said that EGFR vIII does not have any significant role in meningioma. Henceforth, extensive studies are required in this aspect.

## Supporting information

**S1 Raw images.** S1 Fig. EGFR vIII RT-PCR Result (A) Standardized PCR cycle (B) amplification plot showing β-actin (reference material) (C) CT value of assessed samples expressed the beta-actin (reference), result showed that the expression of EFGR vIII was not detected in grade I and grade II meningioma using qRT-PCR. S2 Fig. Screening and mapping of EGFR vIII deletions in MG (A) Design of the RTPCR primers specific for the EGFR vIII mutant. (B)

Design of mapping PCR primers corresponding to the EGFR Viii (C) No DNA sequence results seen. S3 Fig. Western blotting result by Anti-EGFR vIII antibody not detected any band of EGFR vIII in meningioma (Grade I & II)

(PDF)

## Acknowledgments

R.R. thankful to Sir Ganga Ram Hospital, Delhi, India for providing the necessary support. We are thankful to Ms. Parul Chug support to statistical analysis.

## Consent to participate

Informed consent was obtained from all participants included in the study.

## Author Contributions

**Conceptualization:** Rashmi Rana, Dharmendra Kumar Yadav.

**Data curation:** Rashmi Rana, Vaishnavi Rathi, Kirti Chauhan, Kriti Jain, Satnam Singh Chhabra, Rajesh Acharya, Samir Kumar Kalra, Anshul Gupta, Sunila Jain, Dharmendra Kumar Yadav.

**Formal analysis:** Rashmi Rana, Vaishnavi Rathi, Kirti Chauhan, Kriti Jain, Satnam Singh Chhabra, Rajesh Acharya, Samir Kumar Kalra, Anshul Gupta, Sunila Jain.

**Funding acquisition:** Rashmi Rana.

**Investigation:** Samir Kumar Kalra, Dharmendra Kumar Yadav.

**Methodology:** Rashmi Rana, Samir Kumar Kalra, Sunila Jain.

**Resources:** Nirmal Kumar Ganguly.

**Visualization:** Dharmendra Kumar Yadav.

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
