## [Decision Letter · Decision Letter 0]

25 Jun 2021

PONE-D-21-18174

Exploring the role of Epidermal Growth Factor Receptor Variant III in meningeal tumors

PLOS ONE

Dear Dr. Yadav,

Thank you for submitting your manuscript to PLOS ONE. After careful consideration, we feel that it has merit but does not fully meet PLOS ONE’s publication criteria as it currently stands. Therefore, we invite you to submit a revised version of the manuscript that addresses the points raised during the review process.

The manuscript has been reviewed by three independent reviewers. They find merit in this manuscript but have highlighted several areas in which improvement and corrections are necessary. These areas include the organization of the text as well as technical details and must be addressed for the manuscript to be published. Few language issues also need to be resolved by the authors.

We look forward to receiving your revised manuscript.

Kind regards,

Timir Tripathi, Ph.D.

Academic Editor

PLOS ONE

Journal Requirements:

1. Please ensure that your manuscript meets PLOS ONE's style requirements, including those for file naming. The PLOS ONE style templates can be found athttps://journals.plos.org/plosone/s/file?id=wjVg/PLOSOne_formatting_sample_main_body.pdf and https://journals.plos.org/plosone/s/file?id=ba62/PLOSOne_formatting_sample_title_authors_affiliations.pdf

'This study was supported by the Sir Ganga Ram Hospital (SGRH) Delhi, India and funded by

the Indian Council of Medical Research (ICMR), Delhi, India (No.: VIR/24/2020/ECD-I and

LSRB/81/48222/LSRB-369/PEE&BS/2020).

ACKNOWLEDGMENTS'

'NO'

Additional Editor Comments (if provided):

Reviewers' comments:

Reviewer's Responses to Questions

**Comments to the Author**

1. Is the manuscript technically sound, and do the data support the conclusions?

Reviewer #1: Yes

Reviewer #2: Yes

Reviewer #3: Yes

2. Has the statistical analysis been performed appropriately and rigorously? 

Reviewer #1: Yes

Reviewer #2: Yes

Reviewer #3: Yes

3. Have the authors made all data underlying the findings in their manuscript fully available?

Reviewer #1: Yes

Reviewer #2: Yes

Reviewer #3: Yes

4. Is the manuscript presented in an intelligible fashion and written in standard English?

Reviewer #1: Yes

Reviewer #2: Yes

Reviewer #3: Yes

5. Review Comments to the Author

Reviewer #1: This study attempts the confer searches for the position attained by EGFR vIII in progression and expression of meningioma. Immunohistochemistry technique showed that EGFR vIII is highly expressed in benign tumors as compared to the atypical meningioma with a highly significant p-value (p<0.05). The study id good to explore EGFR vIII in Meningioma. In my opinion paper need to minor revision.

1. In page 4, reference no. 10 (Ge at al., 2002), explaining ligand independent activation of EGFR vIII should be elaborated. Further studies could be carried out on this aspect of the mutant.

2. Page 5, line no. 92 there is a grammatical mistake. ‘Levels’ should be in singular form i.e. ‘level’.

3. Method for Ki-67 analysis should be added in the material and method section.

4. Details of primary antibody of EGFR vIII used for immunohistochemistry need to be provided.

5. In the section of DNA analysis, melting temperature (Tm) of primers used should be mentioned.

6. Details of Chemidoc and the software used for the analysis of blots are required to be given.

7. There is a typing error in detail of Fig. 1, ‘her’ should be written as ‘here’.

8. There is a grammatical error in page 5, line no. 100, DNA sequences should be written as ‘DNA sequence’.

Reviewer #2: Rana et. al. reported EGFR vIII in progression and expression of meningioma. . Hence, Opinions regarding the role that EGFR vIII in tumorigenesis and tumor progression are clearly conflicting and, therefore, it is crucial not only to find out its mechanism of action, but also to definitely identify its role in meningioma. This study need to minor revision for publication.

1. Please elaborate the correlation of gender with EGFR vIII in your study.

2. Discuss about the effect of age on the neurological deficit factor of meningioma.

3. In the section material and methods, RT- PCR parameters used should be added.

4. As mentioned in the text, the tissue samples were stored in -80 degree Celsius. Can you elaborate the extent of degradation in the samples?

5. I would suggest the addition of protocol followed for quantitative PCR.

6. Discuss about the duration of exposure with antibodies in western blotting.

7. Under the section results, line 351-352, pg. no. 20 has some grammatical error.

Reviewer #3: Author reported potentialities of the variant in number of tumors but cannot validate the results by conventional methods and further advanced techniques could be helpful in this regard. Moreover, it can also be said that EGFR vIII does not have any significant role in meningioma. This study need minor correction for before publication.

1. In your study, emphasis is given on Ki-67 proliferation. Its significance in this study should be discussed thoroughly.

2. Authors should provide their perspective for non-significant results by different techniques used.

3. Case recurring meningiomas should be discussed by the authors.

4. In the section material and methods, provide the concentration of formalin used for fixation of tissues.

5. Page 23, line 414, ‘targeting’ should be written as ‘targeted’.

6. Under the section discussion, cross-reactivity of antibodies should be discussed briefly.

7. Discuss the future prospects of this study.

6. PLOS authors have the option to publish the peer review history of their article (what does this mean?). If published, this will include your full peer review and any attached files.

Reviewer #1: **Yes: **Abhishek Kumar Singh

Reviewer #2: **Yes: **Yogesh Kumar

Reviewer #3: No

---

## [Author Response · Author response to Decision Letter 0]

8 Jul 2021

Response to Reviewers comments

Title: Exploring the role of Epidermal Growth Factor Receptor Variant III in meningioma

Authors: Rashmi Rana1*, Vaishnavi Rathi1, Kirti Chauhan1, Kriti Jain1, Satnam Singh Chhabra2, Rajesh Acharya2, Samir Kumar Kalra2, Anshul Gupta2, Sunila Jain3, Nirmal Kumar Ganguly1, Dharmendra Kumar Yadav4*

Authors are thankful to the reviewers for their comments which are valuable and helped in improving the content and quality of the article “Exploring the Role of Epidermal Growth Factor Receptor Variant III in meningioma”. In the revised version of the manuscript, corrections are highlighted in blue color for the convenience of reviewers.

Reviewer 1

This study attempts the confer searches for the position attained by EGFR vIII in progression and expression of meningioma. Immunohistochemistry technique showed that EGFR vIII is highly expressed in benign tumors as compared to the atypical meningioma with a highly significant p-value (p<0.05). The study id good to explore EGFR vIII in Meningioma. In my opinion paper need to minor revision.

Response: Authors thankful for appreciation of the work. All the issues related to the manuscript has been resolved and revised the manuscript accordingly. 

Comments: In page 4, reference no. 10 (Ge at al., 2002), explaining ligand independent activation of EGFR vIII should be elaborated. Further studies could be carried out on this aspect of the mutant.

Response. Thank you for valuable suggestion. We made changes according to suggestion of reviewer and error has been corrected in the text.

Comments: Page 5, line no. 92 there is a grammatical mistake. ‘Levels’ should be in singular form i.e. ‘level’.

Response. Thank you for valuable suggestion. We made changes according to suggestion of reviewer and error has been corrected in the text.

Comments: Method for Ki-67 analysis should be added in the material and method section.

Response: Thank you for valuable suggestion. We made changes according to suggestion of reviewer and added in the main text of manuscript.

Comments: Details of primary antibody of EGFR vIII used for immunohistochemistry need to be provided.

Response: We made changes according to suggestion of reviewer and added in main text.

Comments: In the section of DNA analysis, melting temperature (Tm) of primers used should be mentioned.

Response: Thank you for valuable suggestion, DNA analysis, melting temperature (Tm) of primers details have been added in main text.

Comments: Details of Chemi-doc and the software used for the analysis of blots are required to be given.

Response: The Chemi-doc and the software used for the analysis of blots details been added in revised Manuscript.

Comments: There is a typing error in detail of Fig. 1, ‘her’ should be written as ‘here’.

Response: Thank you for valuable suggestion, Error has been corrected in the main text.

Comments: There is a grammatical error in page 5, line no. 100, DNA sequences should be written as ‘DNA sequence’.

Response: Thank you for valuable suggestion, sentences has been changed according to suggestion of reviewer. Error has been corrected in the text. 

Reviewer 2

Rana et. al. reported EGFR vIII in progression and expression of meningioma. . Hence, Opinions regarding the role that EGFR vIII in tumorigenesis and tumor progression are clearly conflicting and, therefore, it is crucial not only to find out its mechanism of action, but also to definitely identify its role in meningioma. This study need to minor revision for publication.

Response: Authors thankful for appreciation of the work. All the issues related to the manuscript has been resolved and revised the manuscript accordingly. 

Comments: Please elaborate the correlation of gender with EGFR vIII in your study.

Response: The study presented here does not the gender of patients. The samples were analyzed irrespective of the sex of the patient. 

Comments: Discuss about the effect of age on the neurological deficit factor of meningioma.

Response: We did not consider the age of the patient. The inclusion criteria for sample collection simply include the meningioma patients with age above 18 years old. Below this age no samples were selected.

Comments: In the section material and methods, RT- PCR parameters used should be added.

Response: Thank you for valuable suggestion, sentences has been changed according to suggestion of reviewer and details have been added in revised MS. 

Comments: As mentioned in the text, the tissue samples were stored in -80 degree Celsius. Can you elaborate the extent of degradation in the samples?

Response: After resection of the tissue, they were stored in -80 degree refrigerator. There is a minimal or no degradation in the sample at this temperature. All the enzymes and metabolic processes occurring in the tissue which leads to its decaying halts at this temperature. Further, tissues were kept in ice when they were taken out for various studies. 

Comments: I would suggest the addition of protocol followed for quantitative PCR.

Response: Thank you for valuable suggestion, sentences has been changed according to suggestion of reviewer and details have been added in revised MS

Comments: Discuss about the duration of exposure with antibodies in western blotting.

Response: Thank you for valuable suggestion, sentences has been changed according to suggestion of reviewer and details discussion has added in revised MS. 

Comments: Under the section results, line 351-352, pg. no. 20 has some grammatical error. 

Response: The error has been corrected in the revised manuscript. 

REVIEWER 3

Author reported potentialities of the variant in number of tumors but cannot validate the results by conventional methods and further advanced techniques could be helpful in this regard. Moreover, it can also be said that EGFR vIII does not have any significant role in meningioma. This study need minor correction for before publication

Response: Authors thankful for appreciation of the work. All the issues related to the manuscript has been resolved and revised the manuscript accordingly. 

Comments: In your study, emphasis is given on Ki-67 proliferation. Its significance in this study should be discussed thoroughly.

Response: Thank you for valuable suggestion, sentences has been changed according to suggestion of reviewer and details of Ki-67 proliferation discussion has added in revised MS. 

Comments: Authors should provide their perspective for non-significant results by different techniques used.

Response: The study presented here, revealed that EGFR vIII expression diminishes as the disease upgrades. This finding came into account through immunohistochemistry analysis. Further, techniques viz. RT PCR, western blotting etc. failed to prove the similar fact. The reason behind this could be the lowered expression of the EGFR vIII thus; it could not be detected by these techniques. The results might appear in the tissue sections but not in the processed forms. 

Comments: Case recurring meningiomas should be discussed by the authors.

Response: The study does not involve the recurring meningioma.

Comments: In the section material and methods, provide the concentration of formalin used for fixation of tissues. 

Response: Thank you for valuable suggestion, sentences has been changed according to suggestion of reviewer and details of concentration of formalin used for fixation of tissues added in main text.

Comments: Page 23, line 414, ‘targeting’ should be written as ‘targeted’.

Response: The error has been corrected in revised manuscript. 

Comments: Under the section discussion, cross-reactivity of antibodies should be discussed briefly.

Response: The cross reactivity faced during the formation of antibodies in other studies have already been discussed in the manuscript. 

Comments: Discuss the future prospects of this study. 

Response: Thank you for valuable suggestion, sentences has been changed according to suggestion of reviewer and details discussion in revised MS. Present study deals with the problems faced in treating meningioma and expression of EGFR vIII in the same. As the disease progress, this EGFR mutant can prove to be a target for treatment or diagnosis of meningioma in future. EGFR vIII shows antagonistic behavior as the disease upgrades. Additionally, research can be done to use this variant in different aspects.

---

## [Decision Letter · Decision Letter 1]

12 Jul 2021

Exploring the role of Epidermal Growth Factor Receptor Variant III in meningeal tumors

PONE-D-21-18174R1

Dear Dr. Yadav,

We’re pleased to inform you that your manuscript has been judged scientifically suitable for publication and will be formally accepted for publication once it meets all outstanding technical requirements.

Kind regards,

Timir Tripathi, Ph.D.

Academic Editor

PLOS ONE

**Additional Editor Comments (optional):**

The reviewers have reviewed the revised manuscript. The authors managed to revise the manuscript as per suggestions and made appropriate changes. Also, they were able to reply to the queries adequately. 

Reviewers' comments:

Reviewer's Responses to Questions

**Comments to the Author**

1. If the authors have adequately addressed your comments raised in a previous round of review and you feel that this manuscript is now acceptable for publication, you may indicate that here to bypass the “Comments to the Author” section, enter your conflict of interest statement in the “Confidential to Editor” section, and submit your "Accept" recommendation.

Reviewer #1: All comments have been addressed

Reviewer #2: All comments have been addressed

2. Is the manuscript technically sound, and do the data support the conclusions?

Reviewer #1: Yes

Reviewer #2: Yes

3. Has the statistical analysis been performed appropriately and rigorously? 

Reviewer #1: Yes

Reviewer #2: Yes

4. Have the authors made all data underlying the findings in their manuscript fully available?

Reviewer #1: Yes

Reviewer #2: Yes

5. Is the manuscript presented in an intelligible fashion and written in standard English?

Reviewer #1: Yes

Reviewer #2: Yes

6. Review Comments to the Author

Reviewer #1: Authors have revised the manuscript adequately. The manuscript may be accepted for publication in its current form.

Reviewer #2: All the comments has been addressed properly in the revised manuscript. I recommend this manuscript for publication.

7. PLOS authors have the option to publish the peer review history of their article (what does this mean?). If published, this will include your full peer review and any attached files.

Reviewer #1: No

Reviewer #2: **Yes: **Yogesh Kumar

---

## [Editor Report · Acceptance letter]

17 Sep 2021

PONE-D-21-18174R1 

Exploring the role of Epidermal Growth Factor Receptor Variant III in meningeal tumors 

Dear Dr. Yadav:

I'm pleased to inform you that your manuscript has been deemed suitable for publication in PLOS ONE. Congratulations! Your manuscript is now with our production department. 

Kind regards, 

on behalf of

Dr. Timir Tripathi 

Academic Editor

PLOS ONE